# Role of Structural and Conformational Diversity for Machine Learning Potentials

**Nikhil Shenoy**[1,2*]    **Prudencio Tossou**[1,3*]    **Emmanuel Noutahi**[1]    **Hadrien Mary**[1]
**Dominique Beaini**[1,4,5]    **Jiarui Ding**[2]

[1]Valence Labs    [2]University of British Columbia
[3]Laval University    [4]Montreal University    [5]MILA
*Equal contribution*

## Abstract

In the field of Machine Learning Interatomic Potentials (MLIPs), understanding the intricate relationship between data biases, specifically conformational and structural diversity, and model generalization is critical in improving the quality of Quantum Mechanics (QM) data generation efforts. We investigate these dynamics through two distinct experiments: a fixed budget one, where the dataset size remains constant, and a fixed molecular set one, which focuses on fixed structural diversity while varying conformational diversity. Our results reveal nuanced patterns in generalization metrics. Notably, for optimal structural and conformational generalization we need a careful balance between structural and conformational diversity that existing QM datasets do not meet. Our results also highlight the limitation of the MLIP models at generalizing beyond their training distribution, emphasizing the importance of defining applicability domain during model deployment. These findings provide valuable insights and guidelines for QM data generation efforts.

## 1 Introduction

Molecular Dynamics (MD) simulations are invaluable tools in the realm of drug and material discoveries. They allow a deeper understanding of the dynamic behavior of biomolecules and materials, shedding light on their structures, functions, and intricate interactions between them and other molecules [18, 37]. For instance, in drug discovery, leveraging MD simulations can improve the estimation of ligand-protein binding energies [19] and kinetics [33, 6, 7, 32]. MDs accuracy and reliability are contingent on the precision of the force fields employed to calculate the changes in energy and forces during the simulations. However, due to their inherent approximations, force fields are not accurate enough and improving them requires a significant expertise and parametrization. Consequently, Machine Learning Interatomic Potentials (MLIPs) trained on Quantum Mechanics (QM) data have emerged as a promising solution to these problems.

MLIPs have gained popularity in the field of atomistic modeling and simulations over the past decade [5, 38, 20, 43, 41, 22, 1, 40]. Their appeal lies in their trade-off between speed and accuracy, enabling expedited calculations while maintaining comparable levels of precision compared to QM methods. They are mainly enabled by the recent developments in ML modeling for physical systems and the creation of large QM datasets that are made publicly available. The first is exemplified by the variety of model architectures and descriptors allowing MLIPs to comprehend the inherent symmetries and biases within atomistic systems and QM modeling [13, 14, 24, 34, 8, 35, 42, 30]. The latter is underscored by the increasing number of efforts to generate and publicly release QM datasets, despite the substantial costs associated with such endeavors [31, 28, 27, 29, 38, 39, 11, 44, 17, 16, 10].

NeurIPS 2023 AI for Science Workshop.

The landscape of MLIP models and their inherent biases, as well as their role in generalization, has received some attention in the recent literature [3], whereas data biases, such as the QM level of theory, the number of labeled molecules and conformers, and the diversity in chemical and conformational aspects, have been comparatively under-explored. These data-specific factors significantly affect the accuracy and generalization capabilities of MLIPs. Consequently, the primary focus of this work is to shed light on the implications of data biases, with the goal of providing valuable insights and guidelines for optimizing the trade-off between the cost of data generation and the value it brings to modeling and generalization efforts.

**Contributions**: First, we designed and conduct experiments to understand the intricate relationship between dataset size, structural diversity, conformational diversity and model generalization. Second, our analysis of generalization is multifaceted allowing the readers to understand how the performance of MLIPs changes within and outside the training distribution of both conformers and structures.

## 2    Related Works

**QM Datasets** Publicly available QM datasets exhibit a wide range of trade-offs between conformational and structural diversity. On one end of the spectrum, we have structurally diverse datasets with no conformational diversity (i.e one conformer per molecule). For instance QM7, QM8, and QM9 [31] respectively comprise 7.1K, 21K, and 133K molecules, each offering only a single energy-minimized conformer per molecule. Larger scale efforts have yielded datasets such as PubchemQC-PM6 [29], PubchemQC-B3LYP/6-31G*//PM6 [27], and Molecule3D [44] which provide a substantial number of molecules—221M, 86M, and 4M, respectively—with a single optimized geometry per molecule and QM properties calculated under various levels of theory.

Moving towards the other end of the spectrum, we have collections with a few molecules but hundreds or thousands of conformers per molecule. For example, QM7X [16] extends the QM7 dataset to encompass 4.2M off-equilibrium conformations for 6.9K molecules. Similarly, DES370K and DES5M [10] consist respectively of 370K and 5M dimer conformations from 400 small molecules, computed at various levels of theory.

In the middle ground, some data collections have both structural and conformational diversity. ANI [38] and its extensions, ANI-1x and ANI-1ccx [39], offer a substantial dataset of 20M off-equilibrium conformations for 57K unique yet diversified molecules, featuring various levels of theory. Likewise, Spice [11] provides a collection of 1.1M conformers for 19K molecules, and GEOM [2], computed using a semi-empirical method, offers 37M energy-optimized conformers for approximately 450K molecules. Meanwhile, QMugs [17] limits itself to three conformers per molecule for 665K drug-like molecules containing up to 100 atoms. Finally, OrbNet Denali [9] contributes 2.3 million equilibrium and off-equilibrium conformers for 200K molecules.

Other aspects of variation among these diverse datasets are presented in Appendix A. Collectively, they illustrate the multifaceted trade-offs, especially between conformational and structural diversity, in the field of QM data generation. They emphasize the critical considerations researchers must make when generating such data or selecting a dataset for training MLIPs.

**Data bias and implications**: Only a couple of studies have delved into the role of QM data biases in model generalization. Glavatskikh et al. [15] contrasted QM9 and PC9 which is a subset of PubChemQC [28], that mimics the size constraints and atom types of QM9 but has greater chemical diversity (meaning herein, higher diversity of functional groups, wider bond length distributions and species with multiplicity $> 1$). The superior generalization of PC9 models suggests that chemical diversity plays a pivotal role in QM model generalization. Frey et al. [12] explored the impact of dataset size on the scaling behavior of invariant GNNs (SchNet [36]) and equivariant GNNs (PaiNN [35] and Allegro [26]). They observed power-law-like scaling behavior in relation to model size, with distinct regimes based on dataset size. Their findings underscore the intricate relationship between dataset size and model complexity in the context of MLIP performance.

Unlike the aforementioned works that concentrate on individual data biases, our study delves into multiple biases, namely dataset size, conformational and structural diversity, and their relationships. We also examine various forms of generalization to provide a comprehensive understanding of MLIP capabilities in the face of changing data biases.

# 3 Method

Let's consider a QM dataset with $N$ datapoints (conformers), encompassing $n_s$ unique molecular structures, with fixed $n_c$ conformers per molecule (i.e $N = n_s \times n_c$). Our investigation seeks to analyze how generalization evolves when altering the dataset size ($N$), the structural diversity ($n_s$), and the conformational diversity ($n_c$). To give a comprehensive picture of MLIPs generalization, we consider four facets of model performance. In the subsequent sections, we will delve deeper into the methodological setup and elaborate on the chosen generalization metrics. It's important to mention that, for the present study, our definition of diversity is primarily based on the count of unique molecules or conformations within a dataset. However, we intend to expand upon this definition in the future to incorporate measures of (dis)similarity as well.

## 3.1 Setup

For our investigation, we run two pivotal experiments, each involving the training of MLIPs on simulated QM datasets characterized by distinct values of $N$, $n_s$, and $n_c$. For a visual representation of these experiments, please refer to Figure 1.

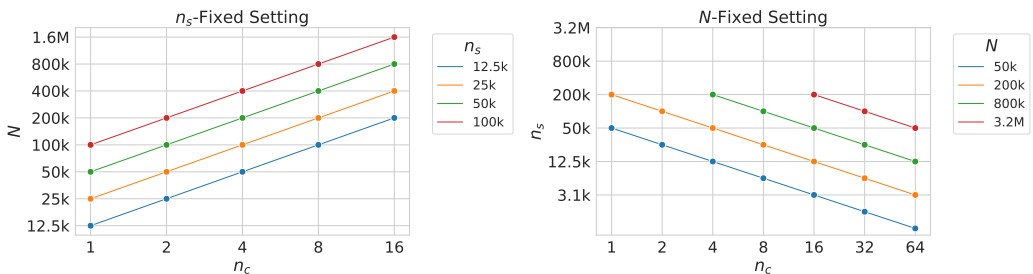

Figure 1: Experimental setup: Left. (1) $n_s$-Fixed: Keeping the number of molecules $n_s$ fixed at 12.5k, 25k, 50k and 100k, we increase the conformer per molecules $n_c$. Right. (2) $N$-fixed: Keeping the total number of conformers $N$ fixed at 50k, 200k, 800k and 3.2M, we increase the conformer per molecule $n_c$ while decreasing the number of molecules $n_s$.

$N$-**fixed experiment**: Herein, we replicate a scenario where there is a fixed budget for data generation. Our objective is to investigate the interplay between structural and conformational diversity and its influence on MLIP generalization. By simulating the generation of QM datasets with a constant number of conformers ($N$), we concurrently vary the values of $n_s$ and $n_c$. Specifically, as $n_s$ decreases, we proportionally increase $n_c$ by the same factor. To illustrate, for $N = 200K$, we generate datasets with ($n_s = 200K, n_c = 1$), ($n_s = 100K, n_c = 2$), ($n_s = 50K, n_c = 4$), ($n_s = 25K, n_c = 8$), and ($n_s = 12.5K, n_c = 16$). This gradual transition spans from a setup featuring low conformational diversity but high structural diversity ($n_s = 200K, n_c = 1$) to one characterized by high conformational diversity and low structural diversity ($n_s = 12.5K, n_c = 16$). By varying $N \in (50K, 200K, 800K, 3.2M)$, our aim is to explore the intricate relationship between this trade-off and the generated dataset size.

$n_s$-**fixed experiment**: This experiment emulates a recent trend in QM data generation, wherein an emphasis is placed on increasing conformational diversity due to its perceived importance in MLIP generalization. Here, our goal is to evaluate the intrinsic impact of conformational diversity on MLIP generalization. To achieve this, we simulate the creation of QM datasets where $n_s$ remains fixed, with values set at $12.5K, 25K, 50K$, and $100K$, while we systematically increase the value of $n_c$ from 1 to 16. The total number of conformers ($N$) is defacto increasing with $n_c$.

$n_c$-**fixed experiment**: We do not conduct any additional experiments for this setting where $n_c$ is fixed while $N$ and $n_s$ increases. To observe the isolated impact of structural diversity on MLIP generalization, we leverage the results from the $N$-fixed and $n_s$-fixed experiments. For instance, for $n_c = 2$, we gather results from experiments where $n_c = 2$ and $N \in [25K, 50K, 100K, 200K]$ from the previous experiments.

## 3.2 Generalization metrics

The distinct aspects of MLIP model performance can be categorized along two axes of generalization. The first axis focuses on the similarity between test samples and the training distribution, distinguishing between samples that are Independent and Identically Distributed (IID) and those that are Out-of-Distribution (OOD). As data points can exhibit variations along both structural and conformational dimensions, the second axis pertains to differentiating chemical characteristics, encompassing both structural and conformational aspects. Consequently, these axes yield four specific generalization metrics for analysis: IID structural (IID-S), OOD structural (OOD-S), IID conformational (IID-C), and OOD conformational (OOD-C).

To calculate the IID-S metric, the test set consists of molecules that share similar physicochemical properties with those in the training set. Conversely, for OOD-S, the test molecules are drawn from a chemical subspace that is distant from the training set. For IID-C and OOD-C metrics, the test sets are composed of novel conformers belonging to molecules encountered during training. To determine whether a conformer is IID-C or OOD-C, we simply compute its minimum Root Mean Square Distance (RMSD) to the training conformers and consider where it falls on that RMSD spectrum. We avoid choosing an arbitrary threshold herein because the spaces of conformers and RMSD are continuous and what is IID or OOD might depend a lot on the molecular energy surface.

# 4 Results

## 4.1 Experimental details

**Datasets:** For our experiments, we use the GEOM dataset [2], a large collection comprising 37 million conformers covering 450K molecules. It has two subsets: GEOM-QM9 made of 133K small molecules from the QM9 dataset [31], with up to 9 heavy atoms (C, N, O, F) and GEOM-Drugs consisting of 317K larger and drug-like molecules. We simulate all our QM data generation by sampling from GEOM-Drugs, and we consider GEOM-QM9 as structurally OOD from it. The structural differences between GEOM-Drugs and GEOM-QM9 are illustrated in Appendix B.

**Model Training:** To train our MLIPs, we use the Equivariant Transformer, a component of the TorchMD-NET models [41]. Our model has approximately 2 million parameters over `num_layers=8` and `hidden_channels=128`. Other hyperparameters are left to their default values [1]. We trained with the L2 loss and the Adam optimizer with a cosine annealing scheduler for the learning rate between $10^{-8}$ and $10^{-4}$.

**Model Evaluation:** We evaluate the models' performance using the mean absolute error (MAE) on the potential energy. The IID-S metric is computed using unseen molecules from GEOM-Drugs and the OOD-S is computed using molecules from GEOM-QM9 as their chemical space is very different from drug-like molecules. IID-C and OOD-C metrics are computed using molecules that have been seen during training according to criteria described in subsection 3.2.

Our experiments are repeated three times using different random seeds, leading to varied data splits and model initializations. For each result, we include error bars to illustrate the standard deviation across these three splits.

## 4.2 Structural generalization

Figure 2 presents the structural generalization metrics for the $N$-fixed experiment, illustrating their dependence on $n_c$ and, implicitly, on $n_s$, as the two variables are inversely related in this setup. Across different values of $N$, we observe a gradual increase in IID-S MAE as $n_c$ increases and $n_s$ decreases. Although the rate of this increase is less pronounced for larger values of $N$, there remains a notable two-fold increase in IID-S MAE when structural diversity decreases by a factor of four and $N = 3.2M$. Conversely, OOD-S MAE also shows an increase with rising values of $n_c$, but these trends are less pronounced across all $N$ values. This phenomenon can partly be attributed to the inherently larger OOD-S MAEs when compared to IID-S MAEs. In fact, the best IID-S MAEs remain in the low single digits, whereas the best OOD-S MAEs hover around $50 kcal/mol$.

---

[1]Implementation as provided in https://github.com/torchmd/torchmd-net

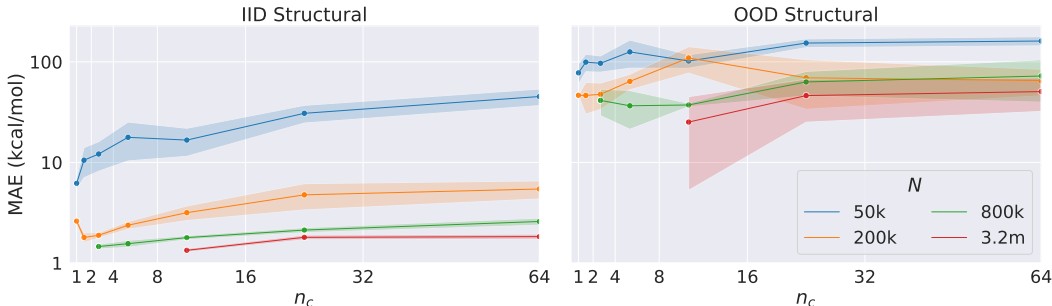

Figure 2: $N$-fixed: Performance on IID-S and OOD-S as we increase the conformational diversity ($n_c$) and reduce structural diversity ($n_s$), while keeping number of conformers ($N$) fixed .

Collectively, these results underscore that within fixed budget constraints, the structural generalization capabilities of MLIPs significantly deteriorate when prioritizing conformational diversity over structural diversity. Consequently, one should exercise caution when opting to sacrifice structural diversity in favor of conformational diversity.

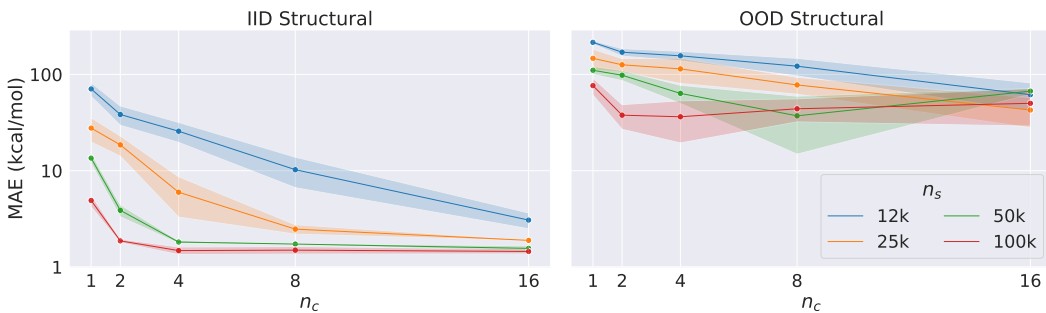

Figure 3: $n_s$-fixed: Performance on IID-S and OOD-S as we increase the conformational diversity ($n_c$) while keeping structural diversity ($n_s$) fixed .

Figure 3 shows the structural generalization metrics for the $n_s$-fixed experiment, demonstrating their dependency on $n_c$ and implicitly on $N$ which are proportional in this setup. For lower values of $n_s$ (i.e., $n_s \in [12K, 25K]$), we observe a gradual reduction in both IID-S and OOD-S MAEs as conformational diversity increases. Although the decrease in MAEs is less pronounced for OOD generalization, it remains notably significant. On the other hand, in cases with higher values of $n_s$ (i.e., $n_s \in [50K, 100K]$), both IID-S and OOD-S MAEs decrease rapidly with small increase in conformational diversity but when it increases further, IID-S MAE plateaus and OOD-S MAE begins to increase. These findings suggest that when structural diversity is low, enhancing conformational diversity can be beneficial. However, as structural diversity increases, the advantages of additional conformational diversity diminish significantly.

Figure 4 shows the structural generalization for the $n_c$-fixed experiment, highlighting the proportional relationship with $N$ and implicitly with $n_s$. A clear trend is observed where increasing the total number of conformers $N$ helps with better IID-S and OOD-S generalization. Additionally, the importance of structural diversity can be observed as experiments with lower $n_c$ or higher $n_s$ generalizes better than the ones with higher $n_c$ or lower $n_s$.

Across both experiments, irrespective of the particular values of $N$, $n_c$, and $n_s$, we consistently observe that IID-S MAEs remain significantly lower than OOD-S MAEs. This emphasizes the MLIP's limited capacity to generalize beyond its training distribution. Therefore, it is imperative for both experimenters and model users to clearly understand the model's structural applicability domain.

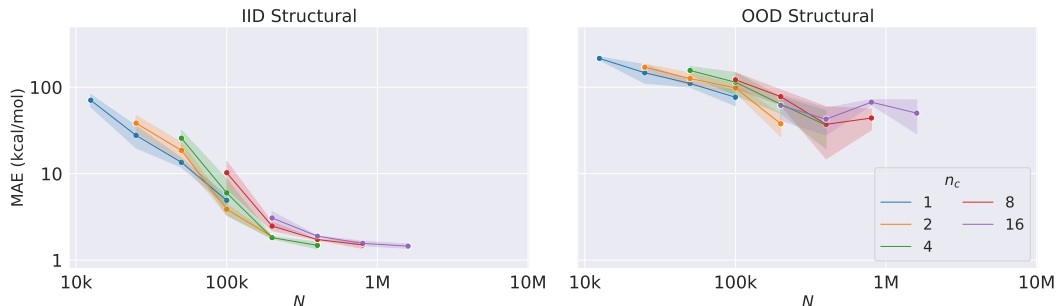

Figure 4: $n_c$-fixed: Performance on IID-S and OOD-S as we increase the number of conformers ($N$) while keeping conformational diversity ($n_c$) fixed .

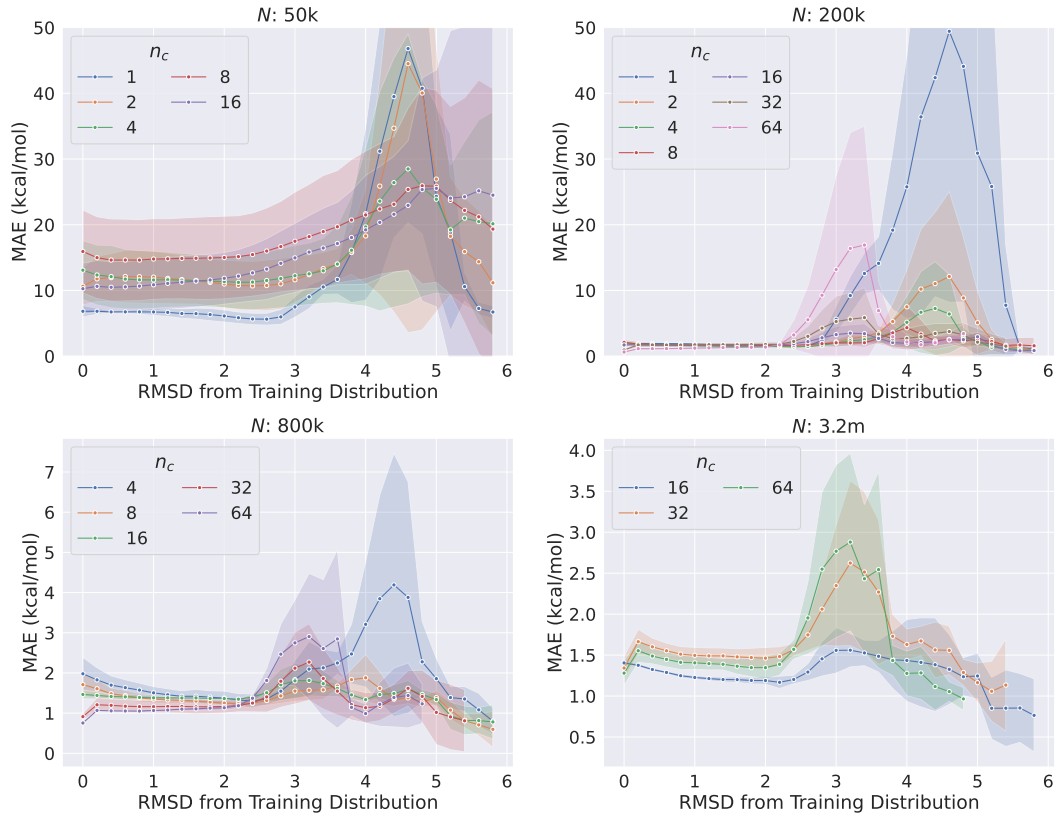

Figure 5: $N$-fixed: Distribution of MAE performance for conformers from both IID-C and OOD-C based on their RMSD from the training distribution. The four plots represent fixed $N$ values (50k, 200k, 800k, and 3.2M) with varying $n_c$ and $n_s$.

## 4.3 Conformational generalization

In Figure 5, we delve into conformational generalization in the $N$-fixed experiment, examining its dependence on $n_c$ (implicitly $n_s$). Across all $N$ values, a consistent pattern emerges: the MAE remains relatively stable when the RMSD to the training conformers is below 2 Å. However, beyond this threshold, we observe an increase in MAE, followed by a return to near-initial values as RMSD continues to increase. Specifically, the plots reveal a steep MAE increase when $3.5$ Å$\leq RMSD \leq$ $5$ Å in scenarios with low conformational diversity ($n_c \leq 4$) but high structural diversity in the training set. Conversely, less steep increases occur when MAE registers between 2.5 Å and 4 Å for high conformational diversity ($n_c \geq 32$) in the training set. The flattest curves are evident when

$n_c \in [8, 16]$, highlighting the need for a delicate trade-off between structural and conformational diversity to achieve effective generalization to unseen conformers of seen molecules.

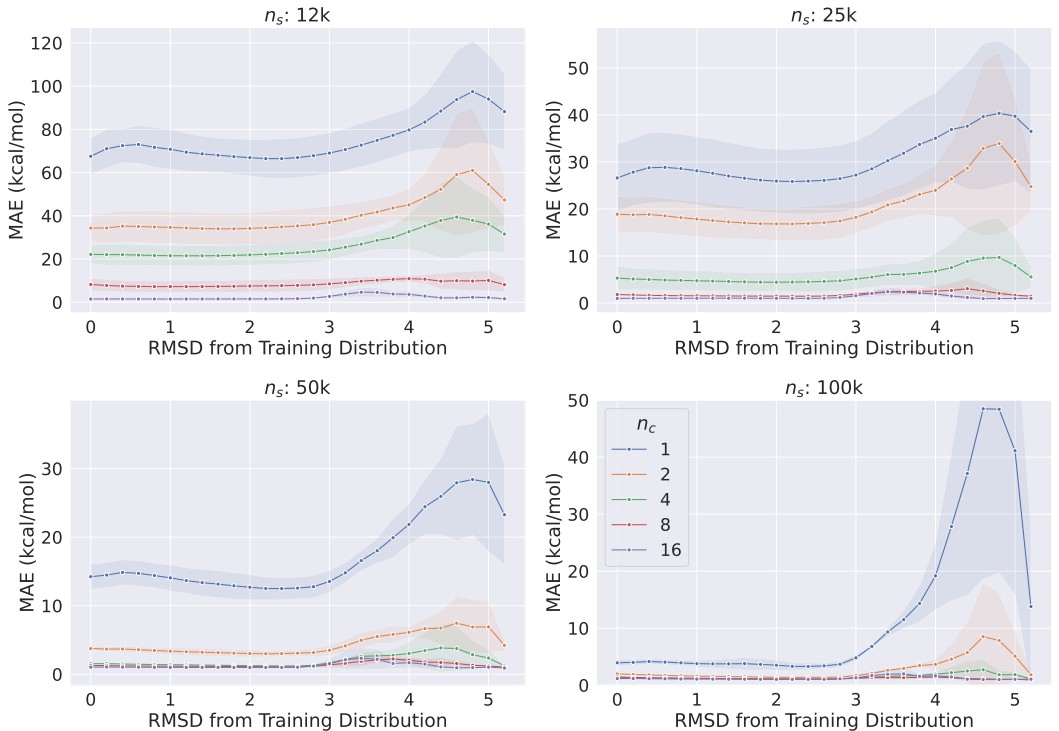

Figure 6: $n_s$-fixed: Distribution of MAE performance for conformers from both IID-C and OOD-C based on their RMSD from the training distribution. The four plots represent fixed $n_s$ values (12k, 25k, 50k, and 100k) with varying $n_c$ and $N$.

In Figure 6, we explore conformational generalization in experiments where structural diversity is fixed, and conformational diversity varies. Across all $n_s$ values, we observe consistent MAE values for all RMSD when $n_c \in [8, 16]$. However, in low conformational diversity settings (i.e., $n_c \leq 4$), MAE remains steady when $RMSD \leq 3$ Å, but as RMSD increases, so does MAE before gradually decreasing. The steepness of these MAE increases and the maximum values reached are inversely related to conformational diversity. This reaffirms the conclusions drawn from the fixed budget experiments: the trade-off between conformational and structural diversity significantly impacts conformational generalization.

In Figure 7, we observe that across all $n_c$ plots, increasing total number of conformers $N$, helps improve conformational generalization across all RMSD values, however the performance gains reduce every time we increase the value of $N$. Additionally, across the different $n_c$ plots, we observe that the maximum MAE observed decreases as $n_c$ increases, suggesting that high $n_c$ is essential for conformational generalization.

While our experiments indicate that the optimal number of conformers per molecule for effective generalization across conformers in both IID and OOD, falls between 8 and 16, it's important to note that this may vary based on other experimental factors such as the network architecture and the chemical space of the training set. Therefore, experimenters should determine the optimal level of conformational diversity tailored to their specific chemical space and MLIP modeling approach.

## 5  Discussion

In the pursuit of developing MLIPs for atomistic modeling, our study delved into the intricate interplay between conformational and structural diversity, data size and model generalization. Through

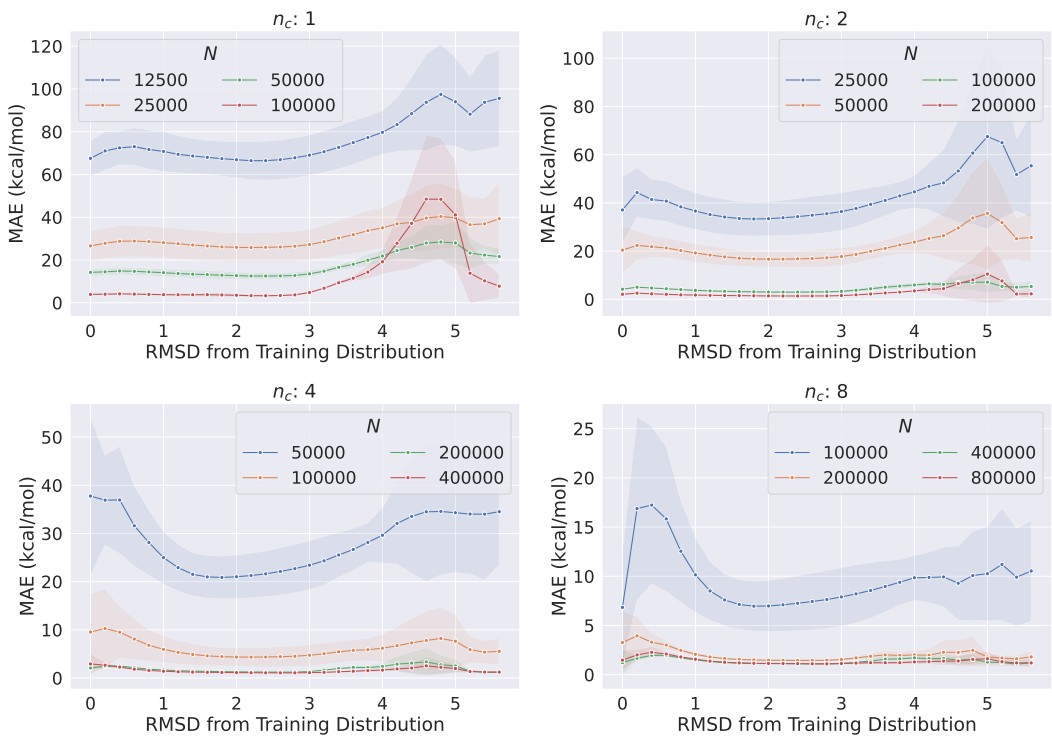

Figure 7: $n_c$-fixed: Distribution of MAE performance for conformers from both IID-C and OOD-C based on their RMSD from the training distribution. The four plots represent fixed $n_c$ values (1, 2, 4, and 8) with varying $n_s$ and $N$.

comprehensive experiments, we unraveled key insights that hold significant implications for the MLIP community.

In the $N$-fixed experiment, where the dataset size remained constant, we discerned that achieving optimal structural generalization necessitates a delicate equilibrium between structural and conformational diversity. The steep rise in MAEs observed when increasing conformational diversity at the expense of structural diversity highlights the need to strike this balance.

Conversely, in the $n_s$-fixed experiment, where structural diversity was kept constant while conformational diversity varied, we observed that the benefits of increased conformational diversity are more pronounced when structural diversity was limited. However, as structural diversity expanded, the advantages of additional conformational diversity diminished, reinforcing the importance of the trade-off.

Throughout both experiments, a consistent pattern emerged: the model's generalization capabilities were constrained within its training distribution, as indicated by substantially lower in-distribution MAEs compared to out-of-distribution MAEs. This underscores the crucial need for researchers and model users to define and recognize the model's applicability domain. Furthermore, the nuanced relationships between conformational and structural diversity and their impact on generalization provide a foundation for future advancements in the field, emphasizing the importance of finding the optimal level of diversity tailored to the specific chemical space and MLIP modeling approach.

While our study has rigorously explored the influence of data biases on MLIP generalization, it uses a specific architecture and dataset, so we acknowledge the need to enhance the validity of our conclusions. Consequently, we intend to conduct a more extensive analysis that encompasses various MLIP modeling biases and incorporates diverse QM datasets. Our plans involve the utilization of alternative QM datasets, employing improved DFT theory levels, incorporating force labels, and leveraging state-of-the-art MLIP architectures, such as Equiformer [23] and MACE [4]. This broader experimentation will provide a comprehensive understanding of the impact of data biases on MLIP generalization, contributing to the advancement of atomistic modeling in various scientific domains.

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

# A  QM-Datasets

Following table lists down the various publicly available QM-Datasets.

Table 1: List of available QM-Datasets and their data generation characteristics

| QM Dataset | Number of Molecules ($n_s$) | Average Conformers per Molecule ($n_c$) | Total Conformers (N) | DFT Theory Level | Atom Types |
|---|---|---|---|---|---|
| GEOM [2] | 450,000 | 82 | 37,000,000 | GFN2-xTB | 18 |
| PubchemQC-PM6 [27] | 221,190,415 | 1 | 221,190,415 | PM6 | 5 |
| PubchemQC- [29] | 85,938,443 | 1 | 85,938,443 | B3LYP/6-31G*//PM6 | 5 |
| Molecule3D [44] | 3,899,647 | 1 | 3,899,647 | B3LYP/6-31G* | 5 |
| NablaDFT [21] | 1,000,000 | 5 | 5,000,000 | $\omega$B97X-D/def2-SVP | 6 |
| QMugs [17] | 665,000 | 3 | 2,000,000 | GFN2-xTB, $\omega$B97X-D/def2-SVP | 10 |
| Spice [11] | 19,238 | 59 | 1,132,808 | $\omega$B97M-D3(BJ)/def2-TZVPPD | 15 |
| ANI [38, 39] | 57,462 | 348 | 20,000,000 | $\omega$B97x:6-31G(d) | 4 |
| DES370K [10] | 3,700 | 100 | 370,000 | CCSD(T) | 20 |
| DES5M [10] | 3,700 | 1351 | 5,000,000 | SNS-MP2 | 20 |
| OrbNet Denali [9] | 212,905 | 11 | 2,3000,000 | GFN1-xTB | 16 |
| QM7-X [16] | 6,970 | 604 | 4,200,000 | PBE0+MBD | 6 |

# B  Structural differences between GEOM-Drugs and GEOM-QM9 Distribution

To illustrate the structural differences between the drug-like molecules from GEOM-Drugs and the small molecules from GEOM-QM9, we create fingerprints for each molecule using the fingerprint function from the datamol library [25]. Subsequently, we extracted two principal components from these fingerprints using Principal Component Analysis (PCA). The resulting principal components were then plotted, revealing a noticeable separation between clusters representing GEOM-Drugs and GEOM-QM9 molecules.

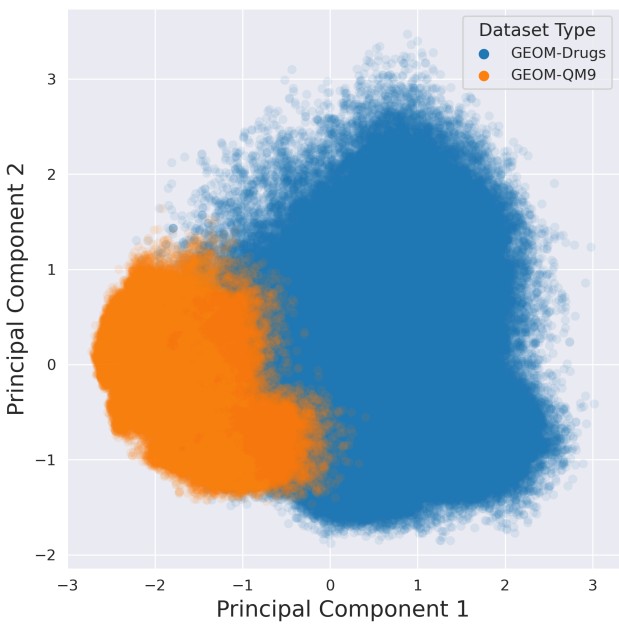

Figure 8: Structural differences between GEOM-Drugs (IID-S set) and GEOM-QM9 (OOD-S set) are evident from the distinct separation between the two clusters. Each point in the plot represents a molecule.

