# OpenReview forum: "Role of Structural and Conformational Diversity for Machine Learning Potentials"
_NeurIPS.cc/2023/Workshop/AI4Science — NeurIPS2023-AI4Science Poster_

### Official Review · Reviewer_JMgx · 2023-10-19
**An important while simple work with excellent writing**

**Rating:** 8
**Confidence:** 4

**Review:**

## Summary
- The authors present a detailed study looking at the interplay between conformational and structural diversity, data size and model generalization. Though the methodology is simple, the finding is important. The writing is excellent and I really enjoy reading it.

## Pros
- An excellent writing with clearly defined motivation and problem set-up, results presentation and discussions. The authors basically give all the information I intent to ask through my reading.

## Cons
- I don't see any obvious flaw of this work. Following are just some suggestions:
- page 2, line 56,  it's better to draw the spectrum explicitly in Figure 1 to help readers better understand it.
- I look forward to seeing the authors future works to incorporate measures of similarity in diversity measurement.
- It is interesting to see the results of Figure 2 for molecules in different size/type classes, I reckon conformational diversity would be more crucial for large and peptide-like biological molecules.

---

### Official Review · Reviewer_nDs4 · 2023-10-22
**Role of Structural and Conformational Diversity for Machine Learning Potentials**

**Rating:** 6
**Confidence:** 3

**Review:**

The authors have constructed a test of testing the variables structural diversity and conformational diversity in constructing a training set for an MLIP. This is an interesting work, and the isolation of the two independent variables is useful to get a sense of the extremes of using largely just different structures and largely just different conformations for a small number of structures.

However I have two key comments:
It is not surprising that when there is more structural diversity, it necessitates for conformations to cover the larger chemical space. The chemical space of interest defines the number of structures needed, and a sufficient ‘threshold’ conformations is needed for sufficient coverage, which is shown by the authors in their between 8 and 16 optimal number of conformers. This is not particularly novel, however it is useful to know what is the threshold number of conformations for different systems. However this threshold number will likely be different for different chemical systems depending on complexity of the chemical space. So I think it should be discussed the generalizability of the conclusions made in this work.
Using the number of structures to represent the structural diversity is not sufficient. How diverse the structures are should be considered - perhaps through Tanimoto similarity or latent space distances or something of the sort. A large number of structures which have similar structures and very different structures would be two different tests.

---

### Meta-Review · Area_Chair_i4xi · 2023-10-27

**Recommendation:** Accept (Oral)
**Confidence:** 3

**Metareview:**

The paper presents a study on the interplay between conformational and structural diversity, data size, and model generalization for machine learning in drug discovery. Reviewers acknowledge the importance of determining the threshold number of conformations for different systems and suggests considering diversity through measures like Tanimoto similarity or latent space distances. Reviewers also appreciate the writing quality and suggests adding an explicit spectrum in Figure 1 and exploring the impact of conformational diversity on different molecule classes.